# Effects of Photobiomodulation Therapy on Performance in Successive Time-to-Exhaustion Cycling Tests: A Randomized Double-Blinded Placebo-Controlled Trial

**DOI:** 10.3390/jfmk8040144

**Published:** 2023-10-11

**Authors:** Fábio Juner Lanferdini, Bruno Manfredini Baroni, Caetano Decian Lazzari, Raphael Luiz Sakugawa, Rodolfo André Dellagrana, Fernando Diefenthaeler, Fabrizio Caputo, Marco Aurélio Vaz

**Affiliations:** 1Biomechanics Laboratory, Centro de Educação Física e Desportos, Universidade Federal de Santa Maria, Santa Maria 97105-900, Rio Grande do Sul, Brazil; 2Physical Therapy Department, Federal University of Health Sciences of Porto Alegre, Porto Alegre 90050-170, Rio Grande do Sul, Brazil; bmbaroni@yahoo.com.br; 3Biomechanics Laboratory, Centro de Educação Física e Desportes, Universidade Federal de Santa Catarina, Florianópolis 88040-900, Santa Catarina, Brazil; caetano.lazzari@gmail.com (C.D.L.); rlsakugawa@gmail.com (R.L.S.); fernando.diefenthaeler@ufsc.br (F.D.); 4Physical Education Department, State University of Ponta Grossa, Ponta Grossa 840030-900, Paraná, Brazil; radellagrana@gmail.com; 5Post-Graduate Program in Movement Sciences, Institute of Health (INISA), Federal University of Mato Grosso do Sul, Campo Grande 79070-900, Mato Grosso do Sul, Brazil; 6Human Performance Research Group, College of Health and Sport Science, Santa Catarina State University, Florianópolis 88080-350, Santa Catarina, Brazil; fabrizio.caputo@udesc.br; 7Exercise Research Laboratory, Escola de Educação Física, Fisioterapia e Dança, Universidade Federal do Rio Grande do Sul, Porto Alegre 90690-200, Rio Grande do Sul, Brazil; marco.vaz@ufrgs.br

**Keywords:** phototherapy, VO_2_ response, muscle oxygenation, cyclists, performance

## Abstract

The goal of this study was to investigate the effects of photobiomodulation therapy (PBMT) on performance, oxygen uptake (VO_2_) kinetics, and lower limb muscle oxygenation during three successive time-to-exhaustions (TTEs) in cyclists. This was a double-blind, randomized, crossover, placebo-controlled trial study. Sixteen cyclists (~23 years) with a cycling training volume of ~460 km/week volunteered for this study. In the first session, cyclists performed a maximal incremental test to determine maximal oxygen uptake and maximal power output (PO_MAX_). In the following sessions, cyclists performed three consecutive TTEs at PO_MAX_. Before each test, PBMT (135 J/thigh) or a placebo (PLA) was applied to both thighs. VO_2_ amplitude, O_2_ deficit, time delay, oxyhemoglobin (O_2_Hb), deoxyhemoglobin (HHb), and total hemoglobin (tHb) were measured during tests on the right vastus lateralis. The PBMT applied before three successive TTE increased performance of the first and second TTE (~10–12%) tests, speed of VO_2_ and HHb kinetics during the first test, and increased peripheral muscle oxygenation (increase in HHb and tHb) in the first and second exhaustion tests. However, the PBMT effects were attenuated in the third TTE, as performance and all the other outcomes were similar to the ones from the PLA intervention. In summary, PBMT application increased the first and second successive TTEs, speed of VO_2,_ and muscle oxygenation.

## 1. Introduction

The skeletal muscles of endurance athletes present greater mitochondrial density (size and volume), which allows for greater energy production through aerobic metabolism [1]. This larger aerobic potential may play an important role in endurance to fatigue, which enables athletes to perform long-lasting training sessions or competitions. In other words, the greater energy availability (via aerobic and/or anaerobic metabolism) in these athletes might delay the onset of fatigue but cannot avoid it. Performance depends on the organism’s ability to produce and deliver energy [2]. However, in high-performance sports, athletes may work at the maximum capacity of their cardiorespiratory and neuromuscular systems. In these high-level sports practices, the energy supply is affected by fatigue and, consequently, performance loss due to the lack of available energy [3]. Therefore, delaying the fatigue processes is one of the factors contributing to performance in some sports, such as cycling.

The deficits between energy supply and exercise demand determine the instauration of the fatigue process and, consequently, the reduction in exercise performance. Therefore, strategies that minimize or delay the deleterious fatigue effects shall contribute to performance increases in endurance sports. Some studies have investigated different acute strategies for improving sports performance [4,5]. A current strategy to delay fatigue effects that were used in recent studies is the application of photobiomodulation therapy (PBMT) over the skeletal muscles prior to exercise since PBMT was able to increase performance during different exercises [6], including running [7,8,9] and cycling [10].

PBMT (low-level laser therapy (LLLT) and/or light emitting diodes therapy (LEDT)) can stimulate mitochondrial metabolism, promoting an increased release of ATP by the local muscle cells and reducing the fatigue effects during strenuous exercises [11]. The mitochondria’s metabolism increase via the PBMT application to the muscle can cause an acceleration in oxygen uptake (VO_2_) kinetics [12] because mitochondria use oxygen as an electron acceptor at the end of the electron transport chain [11,13]. Some studies have investigated the effect of PBMT on VO_2_ [8,12,14,15]. Analysis of the VO_2_ kinetics allows us to determine the efficiency of oxygen delivery by the cardiopulmonary system and the use of oxygen by the mitochondria [16]. Only recently were the effects of the previous application of PBMT on the performance of the time-to-exhaustion (TTE) [10] and VO2 kinetics [12] in competitive cyclists investigated. Authors found an increase in performance with three doses of PBMT (135, 270, and 405 J/thigh) compared to the placebo [10]. In addition, doses of PBMT reduced the oxygen (O_2_) deficit and time constant (Tau) without changes in VO_2_ amplitude [12]. However, in another recent study, changes in VO_2_ kinetics with the previous application of PBMT (540 J/limb) were observed in healthy subjects [17]. Therefore, there seems to exist some uncertainty about the effects of PBMT on VO_2_ kinetic changes.

Despite the numerous investigations into the physiological and/or biomechanical response to successive exhaustion cycling tests and their consequences on performance [18], we were unable to find studies assessing the effects of PBMT application on successive cycling exhaustion tests and on peripheral muscle oxygenation (i.e., using near-infrared spectroscopy—NIRS). NIRS is a technique for monitoring non-invasive muscle oxygenation that allows measurements even in dynamic exercises such as cycling [19]. NIRS is based on the relative ease in which infrared light (700–1000 nm) surpasses biological tissues such as muscle tissue by the amount of light recovered after illumination of the same tissue. Therefore, NIRS allows for the indirect evaluation of mitochondrial function since the process of muscular oxygenation is directly related to the capacity of the muscle to produce energy from the use of greater amounts of oxygen by the mitochondria to generate energy. Increased energy availability (ATP) after PBMT application might be related to a local increase in muscle flow (e.g., increased muscle concentration of hemoglobin). More specifically, the increased ATP concentration in the muscle and plasma due to PBMT may generate vasodilation (ATP is a potent vasodilator) due to ATP’s stimulating effect on the formation of nitric oxide (NO) and prostaglandins, which can offset local sympathetic vasoconstriction [20,21]. This increased muscle blood flow due to vasodilation can be measured via NIRS, which allows the identification of the PBMT action on muscle tissue during exercise in vivo [13]. Thus, additional studies are needed to elucidate the effects of PBMT on cycling performance during successive exhaustion cycling tests and its effects on the use of VO_2_ (e.g., VO_2_ kinetics and peripheral muscle oxygenation).

Therefore, the aim of this study was to evaluate the effects of PBMT on cycling performance, VO_2_ kinetics, and periphery muscle oxygenation during three successive TTEs. We hypothesized that using PBMT before the fatigue tests would increase performance by speeding up VO_2_ kinetics (i.e., lower O_2_ deficit) and increasing muscle oxygen availability.

## 2. Materials and Methods

### 2.1. Experimental Design

Our study was characterized as a crossover, randomized, double-blind trial (blinding of the cyclists and the researcher responsible for evaluations). All protocols were explained to the participants, who voluntarily provided their consent to participate in the investigation through a comprehensible informed consent document. This study received approval from the Research Ethics Committee on Human Subjects at the institution where the research was conducted (number 708.362). The cyclists participating in the present study had ~6.5 years of regular training/competition and no history of lower limb muscle–skeletal injuries. Inclusion criteria included cyclists aged 18–30 years with a competitive history and no history of musculoskeletal injuries in the lower limbs in the last two years. Exclusion criteria included chronic disease, smoking, metabolic disorders, use of steroids in the last six months, chronic disease, physical disabilities, and use of antibiotic drugs in the previous week.

Each cyclist visited the laboratory on three occasions (Figure 1). At the first visit, cyclists performed a maximum incremental test and familiarization to three successive TTEs. At the two subsequent visits, participants performed a standard protocol of three successive tests to exhaustion at maximal power output (PO_MAX_) with preferred cadence, and PBMT or placebo (PLA) treatments were applied before each trial. The three testing days were performed with a 72 h interval apart. A single therapist was tasked with the random allocation of the PBMT and PLA interventions. This therapist received instructions not to disclose the treatment modality employed during each assessment session to either the cyclists or other researchers involved. Furthermore, cyclists utilized opaque eyewear to shield their eyes and obstruct their visual access (visual blinding). The therapist was explicitly advised against revealing the treatment variant to both the cyclists and the overseeing researcher. Notably, PBMT did not elicit any thermal or tactile sensations, thereby ensuring that athletes remained unaware of the specific application on their thigh area. The random assignment transpired through a basic drawing of lots during the first testing session, determining the allocation of either active PBMT or inactive PLA.

### 2.2. Participants

Sixteen male competitive cyclists (age 23 ± 7 years, body mass: 67 ± 7 kg, and height 177 ± 6 cm) participated in this study. Athletes had a training experience of ~6.5 years, a usual training volume of ~6 days/week, and ~460 km/week. G*Power software (Version 3.1.9.6, Franz Faul, Kiel Universität, Kiel, Schleswig–Holstein, Germany) calculated a minimal sample size of 16 subjects (effect size = 0.65; significance level = 0.05; observed power = 0.80) with two-way repeated-measures ANOVA as a statistical test used to compare the conditions and successive TTE.

### 2.3. Procedures

In the first session, anthropometry (height and body mass) was assessed. Subsequently, cyclists engaged in a warm-up routine, sustaining a cycling power output of 150 W for 10 min. The cyclists’ evaluation was conducted utilizing a cycle ergometer (Excalibur Sport, Lode Medical Technology, Groningen, Groninga, The Netherlands) configured to match their bike’s handlebar and seat settings. This setup aimed to determine their maximal power output (PO_MAX_ in Watts). Load increases in the ramp test were performed in 25 W steps every minute (~0.42 W/s) until exhaustion. The cyclists’ preferred cadence was maintained close to 95 ± 5 rpm by providing visual feedback through the cycle ergometer display. After the incremental test, cyclists cycled ~30 min at 50 W for recovery purposes, and, finally, they performed a familiarization in the three successive TTEs at PO_MAX_ and cadence set at 95 ± 5 rpm. During the incremental test, simultaneously with the acquisition of PO, VO_2_ was measured breath-by-breath using an open-circuit gas analyzer (Quark CPET, Cosmed, Rome, Lazio, Italy). Before the incremental test, O_2_ and CO_2_ analyzers were calibrated using medical-grade gases of known concentration that spanned air in the physiological range.

The VO_2_ response obtained during the maximal incremental test was analyzed through visual inspection from the breath-by-breath. VO_2_ measurements were plotted to facilitate the exclusion of values lying beyond four standard deviations above or below the average of the dynamic window (three breaths), taken as the reference for the overall curve average [22]. During the maximal incremental test, PO_MAX_ and VO_2MAX_ values were obtained. VO_2MAX_ was determined as the mean value observed during a 30 s period from the last stage of the test.

Prior to the first TTE, cyclists performed a warm-up with a PO of 150 W for 10 min. After the warm-up, before each consecutive TTE, phototherapy treatment (PBMT or PLA) was performed using a PBMT device (Vectra Genisys Systems, Chattanooga Group, Dallas, TX, USA). The cluster probe consisted of five LLLT diodes (850 nm) and 28 LEDs (670 nm, 880 nm, and 950 nm). PBMT was applied in nine sites of each quadriceps femoris muscle (Figure 2). A dosage of 15 J per site led to a total energy of 135 J per thigh, effectively increasing cycling performance in a previous study [10]. We chose to apply PBMT specifically to the quadriceps femoris because this muscle group is of utmost significance in generating torque and propelling the pedaling cycle. Its pivotal role in cycling performance made it a prime target for the PBMT intervention in our study [23,24,25]. The PLA treatment was performed in exactly the same manner as the PBMT treatment, but with the device switched off, the cluster was held stationary in contact with the skin at a 90° angle, with light pressure on the skin. The total application time of PBMT or PLA was ~5 min for both limbs (9 points per thigh = 18 points × 16 s per point) before each TTE (Figure 2). Following the administration of PBMT or PLA treatments, cyclists underwent a one-minute resting period. Subsequently, they engaged in pedaling at an approximate cadence of ~95 rpm for an additional minute while the cycle ergometer’s resistance remained unloaded (baseline). PBMT or PLA treatments were performed immediately prior to each TTE.

### 2.4. Exhaustive Severe-Intensity Cycling Test Exercise

In sessions 2 and 3, cyclists performed the three successive TTEs at PO_MAX_ at their preferred cadence. Exhaustion was defined as the time when the cyclist was unable to maintain a cadence above 70 rpm. During the successive TTE, cadence and PO were measured using the cycle ergometer software. VO_2_ was measured breath-by-breath using an open-circuit gas analyzer (Quark CPET, Cosmed, Rome, Lazio, Italy), following procedures similar to the ones used in the incremental test. Peripheral muscle oxygenation (Figure 3) data collection was performed using a NIRS (PortaMon, Artinis Medical Systems, Elst, Guéldria, Netherlands) probe (light emitting and photoreceptor), which was positioned in the vastus lateralis muscle belly of the right lower limb, longitudinally located between the femur’s lateral epicondyle and trochanter (~10 cm above the knee joint), fixed with adhesive tape (3M Company, Saint Paul, MN, USA), and wrapped by the cyclist’s bretelle to prevent light penetration during all TTE.

### 2.5. Data Analysis

Analysis of VO_2_ kinetics during successive TTE was converted into 5 s windows, and mean values were calculated. The analysis of VO_2_ kinetics was dependent on the duration of the three TTEs for each athlete in each condition (PBMT or PLA). The first 20 s of the VO_2_ curve (cardiodynamic phase) were excluded from the data analysis for the mono-exponential equation models. Breaths that were aberrant in nature (e.g., arising from actions such as swallowing, coughing, or a lack of identifiable indicators) were omitted from the VO_2_ analysis. This also included the exclusion of VO_2_ values displaying mean values exceeding four times the standard deviation. VO_2_ kinetics were calculated with the nonlinear least-squares method implemented in MATLAB (Mathworks, Natick, MA, USA) to adjust the VO_2_ data. In order to allow the comparison of VO_2_ responses, the data were modeled using the mono-exponential model, isolating the fast component of the VO_2_ curve—Equation (1) [26].

Equation (1):VO_2_(t) = VO_2_ baseline + H (t − TDp) × Ap (1 − exp − ^(t−TDp/τp)^)(1)
where VO_2_ (t) represents the absolute VO_2_ at a given time (t); VO_2_ baseline represents the mean VO_2_ in the baseline period; and Ap, TDp, and p represent the VO_2_ amplitude, time delay, and time constant (Tau = τ), respectively. H represents the Heaviside step function. Total muscular oxygen deficit was also calculated using a descript function (O_2_ Deficit = (τ + TDp) Ap) by Whipp and Casaburi [27].

Data from the NIRS system were analyzed from average windows every 20% of the TTE tests and retests. The oxyhemoglobin (O_2_Hb), deoxyhemoglobin (HHb), and total hemoglobin (tHb) were evaluated. Furthermore, O_2_Hb, HHb, and tHb values were normalized by their respective mean values obtained at rest before the first TTE in each condition, which were used to compare the experimental situations (PBMT vs. PLA). In addition, HHb kinetics were also calculated in all cycling tests until exhaustion. A first-order exponential model fit was used to estimate the parameters tau, amplitude, time delay, and baseline [19].

### 2.6. Statistical Analysis

Data normality and sphericity were evaluated using the Shapiro–Wilk and Mauchly tests, respectively. A two-way repeated-measures ANOVA was used to compare the exhaustion time and kinetics of VO_2_ variables (O_2_ deficit, tau, VO_2_ amplitude, time delay, and baseline), and a three-way ANOVA was used to compare the percentage of muscle oxygenation (O_2_Hb, HHb, and tHb) and HHb kinetics (tau, amplitude, time delay, and baseline) between PBMT and PLA, and between successive TTEs. If the main effects were significant, post-hoc Bonferroni tests were used to identify significant differences. The statistical analysis was performed with an open-source statistical package (JASP 0.11.1 for Windows, Amsterdam, North Holland, Netherlands), with a significance level of α = 0.05. We also calculated the effect size (ES) as proposed by Cohen [28]. All dataset is in Appendix A).

## 3. Results

A flowchart summarizing the procedures of this study is presented in Figure 4. Cyclists presented a PO_MAX_ of 392 ± 31 W and VO_2MAX_ of 67.5 ± 7.5 mL.kg·min^−1^ during the maximal incremental cycling test. Furthermore, cyclists completed the six trials (three in each experimental condition) adequately since the target cadence (100 ± 5 rpm) was maintained during all tests (PLA = test 1 (102 ± 5 rpm); test 2 (103 ± 4 rpm); test 3 (104 ± 4 rpm); and PBMT = test 1 (104 ± 5 rpm); test 2 (103 ± 4 rpm); test 3 (104 ± 4 rpm); condition (*p* = 0.44 and ES = 0.20; time (*p* = 0.135; ES ≤ 0.52)).

Increased cycling performance was observed for test 1 and test 2 with the PBMT application compared to the PLA. PBMT application pre-exercise increased the first TTE by 16.4 ± 10.4 s or 12 ± 15% (*p* < 0.01; ES = 1.58) and test 2 by 11.9 ± 7.0 s or 10 ± 13% (*p* < 0.01; ES = 1.72), without changes in test 3 (4.0 ± 7.6 s; *p* = 0.05; ES = 0.53; Figure 5). Furthermore, the test–retest comparison with the PBMT application showed a reduction in the TTE when comparing test 1 to test 2 (*p* < 0.01; ES = 1.93) and test 1 to test 3 (*p* < 0.01; ES = 2.44). Similarly, PLA application resulted in a reduction in the TTE from test 1 to test 2 (*p* < 0.01; ES = 1.37) and from test 1 to test 3 (*p* < 0.01; ES = 1.73; Figure 5).

PBMT applied prior to the first TTE reduced the O_2_ deficit by 138 ± 183 mL·min^−1^ (*p* < 0.01; ES = 0.82) and tau by 3.5 ± 1.3 s (*p* = 0.01; ES = 0.79) compared to PLA treatment. However, the application of PBMT in the second and third TTE had no effect on VO_2_ kinetics compared to PLA treatment (*p* > 0.05; Table 1).

Furthermore, PBMT and PLA applied before successive TTE showed a reduction of O_2_ deficit (PBMT = test 1 > test 2; *p* < 0.01; ES = 0.96; and test 1 > test 3; *p* < 0.01; ES = 1.39; PLA = test 1 > test 2; *p* < 0.01; ES = 1.57; and test 1 > test 3; *p* < 0.01; ES = 1.48). In addition, PLA applied before successive TTEs showed a reduction in tau between tests (test 1 > test 2; *p* = 0.01; ES = 0.74; and test 1 > test 3; *p* = 0.01; ES = 0.79). Moreover, both conditions showed increased baseline values between successive TTEs (PBMT = test 1 < test 2; *p* < 0.01; ES = 1.23; and test 1 < test 3; *p* < 0.01; ES = 1.20; PLA = test 1 < test 2; *p* < 0.01; ES = 1.22; and test 1 < test 3; *p* < 0.01; ES = 0.76; Table 1).

However, no O_2_Hb changes were observed between PBMT and PLA conditions in any of the tests (*p* > 0.05; Figure 6). Furthermore, a higher HHb concentration was observed with PBMT compared to PLA during the successive TTEs of all sections (20–40–60–80–100%) of test 1 (*p* ≤ 0.01; ES ≥ 0.70) and 2 (*p* ≤ 0.02; ES ≥ 0.65), without alterations for test 3 (*p* > 0.05); Figure 6. In addition, higher concentration of tHb was observed with the use of PBMT compared to PLA during test 1 (20–40–60–80–100%; *p* ≤ 0.04; ES ≥ 0.57) and test 2 (20–40–60–100%; *p* ≤ 0.04; ES ≥ 0.56), without alterations for all sections of test 3 (*p* > 0.05); Figure 6. In addition, PBMT application increased the speed of HHb kinetics (e.g., tau reduction; *p* = 0.04; ES = 1.04) and increased the amplitude of HHb compared to PLA in the first TTE (*p* < 0.05; ES = 0.75), without changes during second and third TTE (*p* > 0.05); Table 2.

## 4. Discussion

According to our findings, PBMT applied to the quadriceps femoris muscles before three successive TTEs increased the cyclists’ performance in the first and second tests. In addition, we observed that PBMT promoted specific changes in the VO_2_ kinetics in the first test and muscular oxygenation (HHb and tHb) in the first and second tests, compared to PLA. However, there was no effect of PBMT for the third test, which indicates an attenuation of the PBMT effects on performance, VO_2_ kinetics, and muscular oxygenation.

Several studies have indicated a notable decrease in fatigue when a pre-fatigue test PBMT is administered, resulting in enhanced endurance performance [8,10]. However, other studies found no performance improvements after PBMT application, especially in anaerobic exercises [15,29]. A possible explanation for these negative results involving PBMT in performance could be the low familiarization of the running or cycling motion. Perhaps another explanation would be that anaerobic exercises would not present differences with the use of PBMT since they do not use the metabolic pathways related to the effect of PBMT on aerobic metabolism, especially through the mitochondrial pathway during the process of muscle fatigue [29].

However, to date, no studies have been found evaluating the effects of the previous application of PBMT on successive TTEs and the corresponding consequences in the performance of athletes familiarized with the cycling motion (i.e., a larger training background). In the present study, the previous application of PBMT resulted in performance increases in the first (12%) and second (10%) successive TTEs, without changes in the third test compared to PLA. These results confirm our hypothesis that PBMT increases endurance performance. In addition, the beneficial effect of PBMT was attenuated throughout the exhaustion tests, possibly due to the accumulation of muscle fatigue processes [3] with the successive TTEs. The reduction in cycling performance during the TTE in both conditions (PBMT and PLA) is especially related to the short time interval adopted between consecutive exhaustion tests (10 min).

As outlined in the existing literature, one of the initial mechanisms by which PBMT affects muscle cells is the augmentation of mitochondrial ATP synthesis through the utilization of the aerobic pathway [11]. According to this mechanism, light energy is absorbed by specific molecular photoacceptors or chromophores, a notable example being cytochrome c-oxidase [13] at the mitochondria, leading to increased rates of ATP synthesis and enhanced mRNA and protein synthesis [11], as well as a shift in the overall cell redox potential and a greater NO release [30]. Consequently, the cascade effects of previous applications of PBMT may lead to improvements in muscle function and, thus, an increase in performance. However, during cycling at successive TTEs with ~3 min of duration, in addition to aerobic metabolism in the energy supply (ATP), there is also the energy supply by the anaerobic metabolism [31]. Therefore, the increased performance with prior application of PBMT could also be due to the higher re-synthesis of Cr-P by mitochondrial shuttle [31], although probably to a lesser extent, because the changes in performance of the third test after PBMT failed to reach statistical significance. In addition, PBMT may modulate several cellular signaling pathways involved in mitochondrial function, such as the AMP-activated protein kinase (AMPK) pathway and the peroxisome proliferator-activated receptor-gamma coactivator 1-alpha (PGC-1α) pathway, both of which play key roles in mitochondrial biogenesis and energy metabolism [32]. The application of PBMT has been extensively studied for its ability to effectively decrease edema and lower markers of oxidative stress and pro-inflammatory cytokines [33]. Moreover, there appears to be a systemic impact whereby light administered to the body can have positive therapeutic benefits on distant tissues and organs [33,34].

The effects of previous high-intensity exercise on subsequent ones per se increase the speed of VO_2_ kinetics and muscle oxygenation in both conditions (PBMT and PLA). It is important to note that the first tests, at which PBMT results on an increased speed of VO_2_ kinetics, were preceded by a 10 min warm-up at moderate intensity (150 W). Such a response following high-intensity, but not moderate-intensity exercises [35], coupled with the decline in performance seen between successive TTE trials due to fatigue-induced metabolic disruptions, might have played a role in diminishing the impact of PBMT across the successive TTE. These results indicate that PBMT and previous high-intensity exercises might share the same mechanistic effects on faster oxidative metabolism. Thus, further studies are necessary to better elucidate the combined effects of different priming exercise intensities (e.g., warm-up) and PBMT on oxidative metabolism and endurance performance.

The results of VO_2_ kinetics and muscle oxygenation also showed attenuation of the effects of the previous application of PBMT throughout the successive TTE. VO_2_ kinetic results showed a reduction in an O_2_ deficit associated with reduced tau at the first TTE, corroborating with previous results on the effects of phototherapy application before cycling TTE, which also found a reduction in the O_2_ deficit and tau [12]. These findings demonstrate that PBMT provoked a reduction in O_2_ deficit and tau, increasing the speed of VO_2_ kinetics compared with the PLA condition. In addition, application of PBMT can increase the activity of complex IV-cytochrome c-oxidase. Likewise, it also elevated the activity of the electrical mitochondrial transport chain complexes [36]. Hence, the enhancement in blood oxygen transportation and subsequent delivery to muscle cells, coupled with the heightened mitochondrial metabolism, can modulate the O_2_ response during successive TTE, explaining the VO_2_ kinetic results.

Muscle oxygenation at the vastus lateralis showed a higher concentration of HHb and tHb during exercise, without changes in O_2_Hb with the previous application of PBMT in the first and second cycling successive TTEs compared to the PLA situation. Any change in HHb reflects the balance between O_2_ delivery and O_2_ utilization in the microvasculature of the muscle region being exercised. Although no study found PBMT effects on muscle oxygenation in athletes, some inferences can be made about the effects of previous applications of PBMT on peripheral muscle oxygenation. The observed increase in HHb, without changes in O_2_Hb during the first and second TTE, can be associated with an increased need for oxygen tissue extraction (e.g., speeding of HHb kinetics) to meet the increased metabolic demands during exercise [37] and with increased muscle blood flow (e.g., increase in tHb after PBMT [13]). However, an enhanced muscle O_2_ supply might also result in slower [HHb] kinetics, as previously reported [38]. Nevertheless, this effect might be difficult or impossible to discern if PBMT also enhances muscle O_2_ extraction, as evidenced by increased mitochondrial activity, providing higher levels of cellular respiration and resynthesizing of ATP [11,30,36] in animal studies. Furthermore, studies in humans have found improvements in muscle function and fatigue resistance, increased oxygen consumption during maximal exhaustion tests [8,31], reduction in O_2_ deficit and tau during the TTE cycling test [12], increased muscle oxygenation [13], reduced blood lactate concentration, and muscle damage markers with previous application of PBMT [8,31]. Therefore, the increase in ATP synthesis, RNA expression, and protein synthesis leads to changes in intracellular redox potential and increased NO release [30]. These biochemical cascade reactions induced via PBMT can lead to improved muscle function and, consequently, an increase in performance. Moreover, additional research is needed to identify the optimal settings for PBMT treatment and to understand how these settings impact the physiological muscle–skeletal response to exercise and sports performance. Hence, future studies examining the effects of PBMT on sports performance, recovery markers, delayed-onset muscle soreness, and perceived recovery could be valuable and of interest to both the scientific community and the general public [6,39].

### 4.1. Limitations

An important aspect of the present study was the analysis of VO_2_ and HHb kinetics using a single transition. While “similar” exercise bouts have traditionally been averaged to improve the data signal-to-noise ratio, in the present study, we were not able to average the exercise transitions due to noticeable effects of the first maximal test on the second TTE and of the second test on the third test in both conditions (PBMT and PLA). We decided on an incomplete recovery between trials to better analyze the effects of PBMT on successive TTE maximal cycling exercises. Therefore, only one testing day was conducted for each condition because of the extremely demanding nature of this exercise. However, it has been observed that the uncertainty of the VO_2_ kinetics tau is inversely proportional to the amplitude of the VO_2_ kinetics, and severe intensity exercises have the highest VO_2_ kinetics amplitude [40]. For this reason, in the present study, the data signal-to-noise ratio might be considered acceptable [41], improving the confidence in the tau values observed herein. Our study did not investigate the use of PBMT on other muscle groups (e.g., gluteus maximus and plantar flexors), which play important roles in cycling performance [42]. As a result, the potential benefits of PBMT on successive TTEs and physiological variables might have been underestimated. Another possible limitation of the study is that we did not use a control group to verify if the baseline VO_2_ kinetics measurements had similar behavior to those obtained with the PBMT and PLA situations.

Furthermore, future studies should measure neuromuscular, physiological, and perhaps biochemical marker changes during three successive TTEs to better determine the exact fatigue effects that impair cycling performance. It is also important to highlight that future studies may also want to verify the PBMT effects on performance during time trials or in other protocols with workload oscillations. This will allow us to better understand the mechanisms of PBMT action on muscular performance during road cycling, mountain biking, and track cycling.

### 4.2. Practical Applications

Our results suggest that PBMT might be a useful ergogenic method to promote increases in endurance performance in competitive cyclists. Therefore, its application in high-intensity endurance interval training could also help cyclists increase the training load/intensity, which may translate into increased endurance training adaptations. While there is some evidence supporting the effectiveness of PBMT in certain exercise scenarios, the precise influence of PBMT on biological activities in humans, especially in skeletal muscle, has not been fully elucidated [39].

## 5. Conclusions

In summary, the aim of our study sought to examine the impact of PBMT on cycling performance, VO_2_ kinetics, and vastus lateralis muscle oxygenation during three consecutive TTEs. Our findings reveal that PBMT application before three successive TTEs increased the performance of the first and second tests (~10–12%), increased the speed of VO_2_ and HHb kinetics during the first test, and increased peripheral vastus lateralis muscle oxygenation (increase in HHb and tHb) in the first and second TTE. However, there appears to be an attenuation of the PBMT effects throughout the tests that seem to be related to high-intensity priming exercise, indicating that they could share the same mechanisms. The use of PBMT appears to yield beneficial effects by safeguarding and enhancing performance during TTE cycling. These findings strongly indicate that PBMT holds promise as a valuable clinical intervention for promoting recovery and enhancing overall sports performance.

## Figures and Tables

**Figure 1 jfmk-08-00144-f001:**
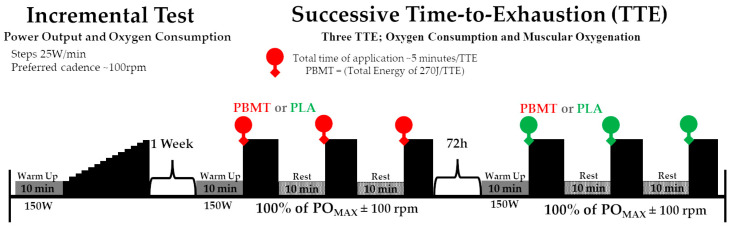
Experimental design. PBMT: photobiomodulation therapy; PLA: placebo.

**Figure 2 jfmk-08-00144-f002:**
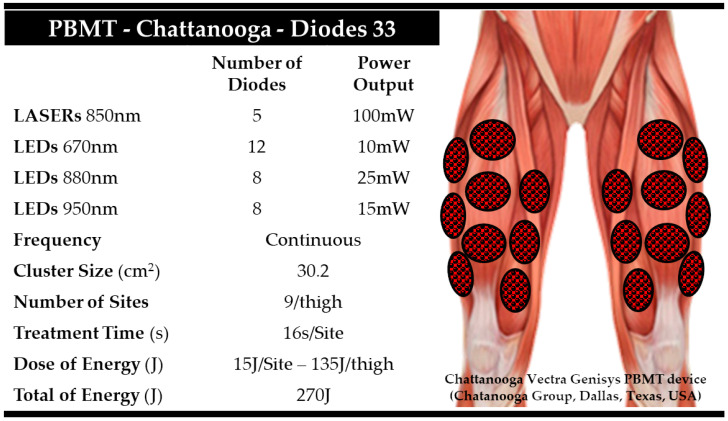
Photobiomodulation therapy (PBMT) parameters and application sites of phototherapy or placebo prior to maximal intermittent exhaustion tests, as well as the PBMT cluster device used and illustration of application points in each thigh of each cyclist.

**Figure 3 jfmk-08-00144-f003:**
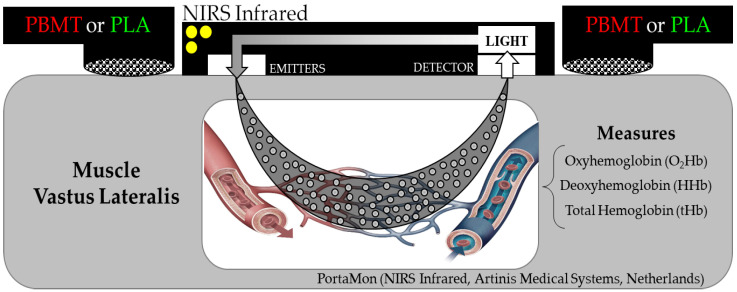
Schematic demonstration of how the NIRS infrared system measures the oxyhemoglobin (O_2_Hb), deoxyhemoglobin (HHb), and total hemoglobin (tHb) variables in the right vastus lateralis muscle during three successive time-to-exhaustion (TTE) cycling test with an application of photobiomodulation therapy (PBMT) and placebo (PLA).

**Figure 4 jfmk-08-00144-f004:**
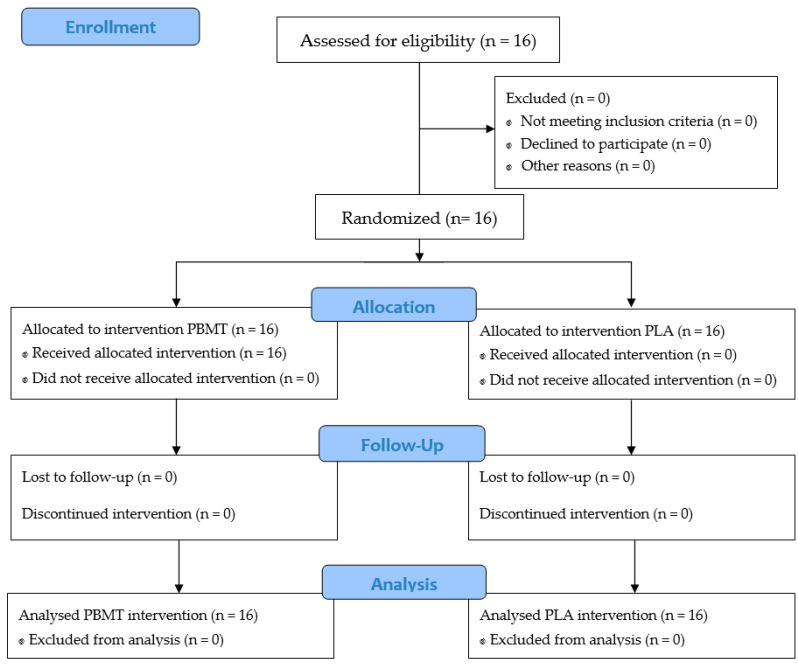
CONSORT flowchart. Placebo (PLA); photobiomodulation therapy (PBMT).

**Figure 5 jfmk-08-00144-f005:**
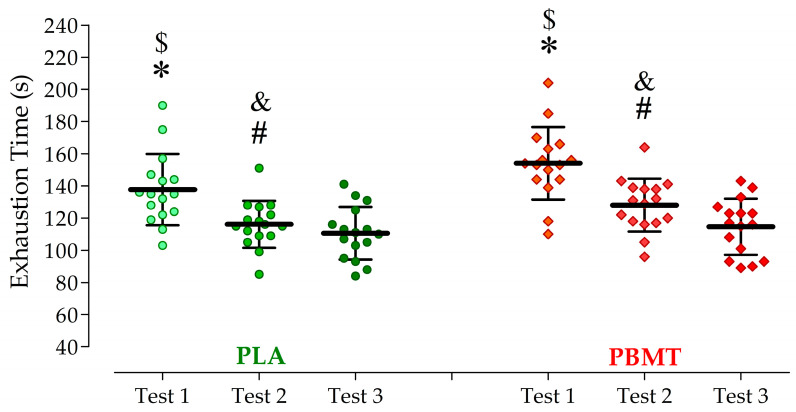
Results of cycling time-to-exhaustion (TTE) during three maximal intermittent cycling tests after application of a placebo (PLA) or photobiomodulation therapy (PBMT). * differences between PBMT and PLA in the first test (*p* < 0.01; effect size—ES = 1.58). # differences between PBMT and PLA in the second test (*p* < 0.01; ES = 1.72). $ differences in TTE between test 1 and test 3 in both conditions (*p* < 0.05; ES > 0.80). & = differences in TTE between test 2 and test 3 in both conditions (*p* < 0.05; ES > 0.80).

**Figure 6 jfmk-08-00144-f006:**
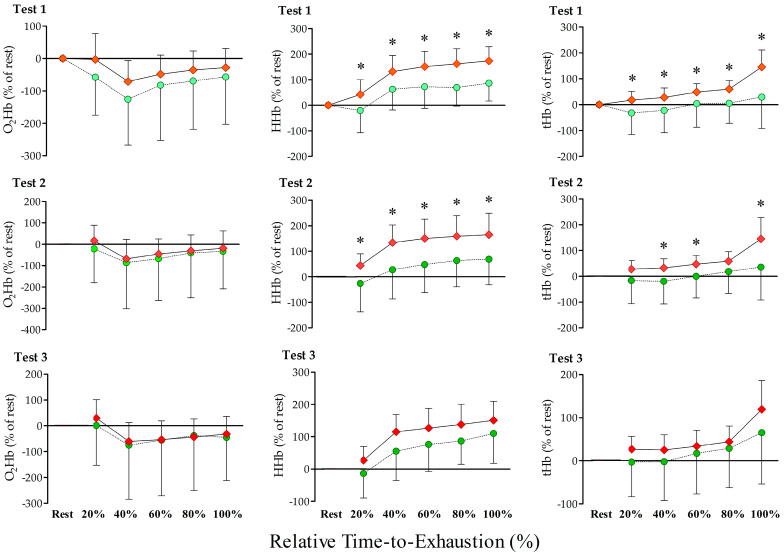
Comparison between placebo (PLA (dashed line and green circles)) versus photobiomodulation therapy (PBMT (solid line and red diamond)) results for peripheral muscle oxygenation normalized for rest (oxyhemoglobin (O_2_Hb), deoxyhemoglobin (HHb), and total hemoglobin (tHb)), during three successive time-to-exhaustions (TTE). * Significant differences between PBMT and PLA in the first and second exhaustion tests (*p* < 0.05).

**Table 1 jfmk-08-00144-t001:** VO_2_ kinetics during three successive time-to-exhaustions (TTEs) in two situations (photobiomodulation therapy (PBMT) and placebo (PLA))—mean ± SD. Significant differences for *p* < 0.05 and effect size—ES.

		PBMT		PLA		PBMT vs. PLA	
Test 1						*p*-Value	ES
Baseline (mL·min^−1^)		531 ± 59		549 ± 52		0.888	0.27
Time Delay (s)		6.0 ± 2.7		5.5 ± 2.5		0.226	0.14
Tau (s)		13.9 ± 3.6		17.4 ± 4.3		0.007	0.79
Amplitude of VO_2_ (mL·min^−1^)		3803 ± 343		3825 ± 401		0.737	0.10
O_2_ Deficit (mL·min^−1^)		783 ± 167		921 ± 171		0.005	0.82
**Test 2**							
Baseline (mL·min^−1^)		635 ± 100		631 ± 91		0.888	0.04
Time Delay (s)		6.0 ± 1.8		5.1 ± 2.1		0.226	0.43
Tau (s)		12.5 ± 2.2		14.2 ± 3.3		1.000	0.48
Amplitude of VO_2_ (mL·min^−1^)		3808 ± 360		3826 ± 366		0.737	0.05
O_2_ Deficit (mL·min^−1^)		659 ± 119		724 ± 94		1.000	0.48
**Test 3**							
Baseline (mL·min^−1^)		618 ± 79		613 ± 102		0.888	0.05
Time Delay (s)		5.5 ± 2.3		5.2 ± 2.3		0.226	0.11
Tau (s)		12.0 ± 2.4		13.4 ± 2.0		1.000	0.52
Amplitude of VO_2_ (mL·min^−1^)		3743 ± 295		3758 ± 243		0.737	0.03
O_2_ Deficit (mL·min^−1^)		630 ± 108		670 ± 130		1.000	0.27
		Test 1 vs. 2		Test 1 vs. 3		Test 2 vs. 3	
		*p*-value	ES	*p*-value	ES	*p*-value	ES
Baseline (mL·min^−1^)	PBMT	0.001	1.23	0.001	1.20	1.000	0.21
	PLA	0.001	1.22	0.001	0.76	1.000	0.43
Time Delay (s)	PBMT	0.685	0.01	0.685	0.17	0.685	0.22
	PLA	0.685	0.16	0.685	0.09	0.685	0.06
Tau (s)	PBMT	1.000	0.43	0.544	0.68	1.000	0.20
	PLA	0.008	0.74	0.001	0.79	1.000	0.32
Amplitude of VO_2_ (mL·min^−1^)	PBMT	0.730	0.01	0.730	0.11	0.730	0.13
	PLA	0.730	0.00	0.730	0.16	0.730	0.21
O_2_ Deficit (mL.min^−1^)	PBMT	0.003	0.96	0.001	1.39	1.000	0.28
	PLA	0.001	1.57	0.001	1.48	1.000	0.50

Amplitude of VO_2_: amplitude of oxygen uptake; O_2_ deficit: oxygen deficit.

**Table 2 jfmk-08-00144-t002:** Deoxyhemoglobin (HHb) kinetics during three successive maximal time-to-exhaustions (TTEs) in two situations (photobiomodulation therapy (PBMT) and placebo (PLA))—mean ± SD between PBMT and PLA. Significant differences for *p* < 0.05.

		PBMT		PLA		PBMT vs. PLA	
Test 1						*p*-Value	ES
Baseline (µM)		−2.5 ± 3.2		−4.4 ± 3.7		0.385	0.37
Time Delay (s)		5.2 ± 1.7		5.9 ± 1.7		0.218	0.46
Tau (s)		9.4 ± 2.4		11.8 ± 2.3		0.039	1.04
Amplitude (µM)		14.6 ± 3.4		11.9 ± 1.6		0.049	0.75
**Test 2**							
Baseline (µM)		−2.0 ± 2.6		−3.3 ± 5.0		0.385	0.21
Time Delay (s)		5.2 ± 1.0		5.2 ± 1.7		0.218	0.03
Tau (s)		10.4 ± 2.3		11.9 ± 4.7		1.000	0.42
Amplitude (µM)		13.9 ± 3.7		11.9 ± 3.2		0.293	0.57
**Test 3**							
Baseline (µM)		−1.8 ± 3.2		−2.5 ± 6.0		0.385	0.10
Time Delay (s)		5.2 ± 1.8		5.8 ± 2.8		0.218	0.19
Tau (s)		12.0 ± 4.1		12.2 ± 4.9		1.000	0.08
Amplitude (µM)		12.6 ± 3.7		11.2 ± 4.2		0.694	0.35
		Test 1 vs. 2		Test 1 vs. 3		Test 2 vs. 3	
		*p*-value	ES	*p*-value	ES	*p*-value	ES
Baseline (µM)	PBMT	0.979	0.28	0.870	0.32	0.999	0.14
	PLA	0.628	0.44	0.083	0.49	0.847	0.24
Time Delay (s)	PBMT	0.724	0.02	0.724	0.00	0.724	0.03
	PLA	0.724	0.36	0.724	0.02	0.724	0.19
Tau (s)	PBMT	0.314	0.42	0.314	0.57	0.314	0.37
	PLA	0.314	0.02	0.314	0.08	0.314	0.07
Amplitude (µM)	PBMT	0.068	0.33	0.068	0.75	0.068	0.76
	PLA	0.068	0.01	0.068	0.18	0.068	0.18

ES: effect size.

## Data Availability

The data that support the outcomes of the present study are available online at FigShare since 9 October 2023, see the link: https://doi.org/10.6084/m9.figshare.24274819.

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
