# Peer review of "Effects of Photobiomodulation Therapy on Performance in Successive Time-to-Exhaustion Cycling Tests: A Randomized Double-Blinded Placebo-Controlled Trial"

_jfmk, 2023, doi:10.3390/jfmk8040144_

Round 1
Reviewer 1 Report
The scientific paper "Effects of Photobiomodulation Therapy on Performance in Successive Time-to-Exhaustion Cycling Tests: A Randomized Double-Blinded Placebo-Controlled Trial" aimed to investigate the effects of photobiomodulation therapy (PBMT) on performance, oxygen uptake (VO2) kinetics, and lower limb muscle oxygenation during three successive time-to-exhaustion (TTE) in cyclists.
The manuscript contains an interesting clinical study. It can be considered that:
1) In the abstract, in the first insertion of the acronym HHb insert the full meaning of the abbreviation;
2) Was the study registered at REBEC or did it use Consort? If yes, mention in the methodology;
3) Insert in the text clearly the inclusion and exclusion criteria of the participants;
4) Was the study developed during the COVID-19 pandemic? Did the cyclists wear masks that might alter the results? Please clarify.
5) In the manufacturers of the materials used, when they are from the United States, insert the state in which it is located. Example PBMT device;
6) Preferably use the full name of the quadriceps muscle: quadriceps femoris;
7) In the manufacturers of the materials used, insert the city in which it is located (manufacturer, city, country);
8) Figures should be colored for better visualization;
9) In the footer of the tables, insert the abbreviations used in it and their meaning.
10) For a clinical study, the number of bibliographic references should be close to 50 articles. Perhaps deepen the discussion on the relationship between PBMT and nerve injuries that affect the muscles (Ex: 10.3390/ph16050653).
11) In the conclusions, initially insert a brief context of the objectives and findings of the research, as well as ending with the benefits and clinical applicability.
Minor editing of English language.
Author Response
Reply to Reviewers’ Comments
Article type: Original Research
Manuscript title: EFFECTS OF PHOTOBIOMODULATION THERAPY ON PERFORMANCE IN SUCCESSIVE TIME-TO-EXHAUSTION CYCLING TESTS: A RANDOMIZED DOUBLE-BLINDED PLACEBO-CONTROLLED TRIAL
Reply to reviewers’ comments
The authors would like to thank the Reviewers for the constructive comments, criticism, and feedback. All the reviewers’ comments have been addressed. In addition, edits to the revised manuscript have been added as tracked changes in the text.
REVIEWER 1
The scientific paper "Effects of Photobiomodulation Therapy on Performance in Successive Time-to-Exhaustion Cycling Tests: A Randomized Double-Blinded Placebo-Controlled Trial" aimed to investigate the effects of photobiomodulation therapy (PBMT) on performance, oxygen uptake (VO2) kinetics, and lower limb muscle oxygenation during three successive time-to-exhaustion (TTE) in cyclists.
The manuscript contains an interesting clinical study. It can be considered that:
- In the abstract, in the first insertion of the acronym HHb insert the full meaning of the abbreviation.
Response: Thank you for your observation. Text has been modified as requested.
- Was the study registered at REBEC or did it use Consort? If yes, mention in the methodology.
Response: Thank you for your observation. Registration has not yet taken place.
- Insert in the text clearly the inclusion and exclusion criteria of the participants.
Response: Thank you for your comment. Such suggestions are relevant and were included in the manuscript (Lines 112-118).
“The cyclists participating in the present study had ~6.5 years of regular training/competition and no history of lower limb muscle skeletal injuries. Inclusion criteria included cyclists with 18-30 years, with a competitive history and no history of musculoskeletal injuries in the lower limbs in the last two years. Exclusion criteria included chronic disease, smoking, metabolic disorders, use of steroids in the last six months, chronic disease, physical disabilities, smoking, and use of antibiotic drugs in the previous week.”
- Was the study developed during the COVID-19 pandemic? Did the cyclists wear masks that might alter the results? Please clarify.
Response: Thank you for your comment. The data collection of the present manuscript were carried out between the years 2015 and 2018. Therefore, the pandemic of COVID-19 had no effect on the results.
- In the manufacturers of the materials used, when they are from the United States, insert the state in which it is located. Example PBMT device.
Response: Thank you for your comment, all changes in text have been made.
- Preferably use the full name of the quadriceps muscle: quadriceps femoris.
Response: Thank you for your comment.
All changes in text have been made by inserting the full name “quadriceps femoris”.
- In the manufacturers of the materials used, insert the city in which it is located (manufacturer, city, country).
Response: Thank you for your comment, all changes in text have been made.
8) Figures should be colored for better visualization.
Response: Thank you for your comment. All figures have been changed to color.
9) In the footer of the tables, insert the abbreviations used in it and their meaning.
Response: Thank you for your comment. All tables and figures have been changed according to the comment.
10) For a clinical study, the number of bibliographic references should be close to 50 articles. Perhaps deepen the discussion on the relationship between PBMT and nerve injuries that affect the muscles (Ex: 10.3390/ph16050653).
Response: Thank you for your comment. Changes to the text were made to adjust the suggestions. Discussion (Lines: 369-376; 422-428); limitations (Lines: 442-445); Practical Application (Lines: 460-463) and in Conclusion.
11) In the conclusions, initially insert a brief context of the objectives and findings of the research, as well as ending with the benefits and clinical applicability.
Response: Thank you for your comment. The conclusion of the manuscript was adequate according to the commentary.
REFERENCES
Coyle, E.F.; Feltner, M.E.; Kautz, S.A.; Hamilton, M.T.; Montain, S.J.; Baylor, A.M.; Abraham, L.D.; Petrek, G.W. Physiological and biomechanical factors associated with elite endurance cycling performance. Med. Sci. Sport. Exerc. 1991, 23, 93–107.
Diefenthaeler, F.; Coyle, E.F.; Bini, R.B.; Carpes, F.P.; Vaz, M.A. Muscle activity and pedal force profile of triathletes during cycling to exhaustion. Sports Biomech. 2012, 11, 10-19.
Kordi, M.; Folland, J.; Goodall, S.; Haralabidis, N.; Maden-Wilkinson, T.; Sarika Patel, T.; Leeder, J.; Barratt, P.; Howatson, G. Mechanical and morphological determinants of peak power output in elite cyclists. Scan. J. Med. Sci. Sport. 2020, 30, 227–237.
Bathini, M.; Raghushaker, C.R.; Mahato, K.K. The molecular mechanisms of action of photobiomodulation against neurodegenerative diseases: a systematic review. Cell. Mol. Neurobiol. 2022, 42, 955-971.
Hamblin, M.R. Mechanisms and applications of the anti-inflammatory effects of photobiomodulation. AIMS. Biophys. 2017, 4, 337-361.
Moskvin, S.V.; Khadartsev, A.A. Methods of effective low-level laser therapy in the treatment of patients with bronchial asthma (literature review). Biomedicine. 2020, 28, 10, 1-20.
Dutra, Y.M.; Malta, E.S.; Elias, A.S.; Broatch, J.R.; Zagatto, A.M. Deconstructing the ergogenic effects of photobiomodulation: a systematic review and meta‑analysis of its efficacy in improving mode‑specific exercise performance in humans. Sports Med. 2022, 52, 2733-2757.
Elmer, S.J.; Barratt, P.R.; Korff, T.; Martin, J.C. Joint-specific power production during submaximal and maximal cycling. Med. Sci. Sport. Exerc. 2011, 43, 1940–1947.
Reviewer 2 Report
There is no justification for the PBM methodology:
- why these particular zones are chosen;
- why 9 zones and not 3-4-5...18, for example?
- why these wavelengths and other parameters are chosen? Just because such a device was available?
- There is no such thing as dose (see publication)! Moskvin S.V. Response to: Optimization of Photobiomodulation Protocol for Chemotherapy-Induced Mucositis in Pediatric Patients (re: doi: 10.1089/photob.2019.4794) // Photobiomodulation, Photomedicine, and Laser Surgery. - 2020; 38 (11): 703. doi: 10.1089/photob.2020.4934.
An important aspect such as the corresponding changes in indicators outside the illumination zone, in other muscles, is not considered.
The fact that there is a direct effect of PBM on blood during out-of-muscle illumination is ignored (see publications below). Focusing attention only on muscles leads to erroneous conclusions regarding the mechanisms of biomodulating action of PBM.
Moskvin S.V., Khadartsev A.A. Methods of effective low-level laser therapy in the treatment of patients with bronchial asthma // BioMedicine. - 2020; 10 (1): 1-20. doi: 10.37796/2211-8039.1000.
Moskvin S.V., Kochetkov A.V. Russian low level laser therapy techniques for brain disorders // Photobiomodulation in the Brain. Low-Level Laser (Light) Therapy in Neurology and Neuroscience / M.R. Hamblin, Y.-Y. Huang (Eds). - London: Academic Press is an imprint of Elsevier, 2019. - P. 545-572.
Author Response
Reply to Reviewers’ Comments
Article type: Original Research
Manuscript title: EFFECTS OF PHOTOBIOMODULATION THERAPY ON PERFORMANCE IN SUCCESSIVE TIME-TO-EXHAUSTION CYCLING TESTS: A RANDOMIZED DOUBLE-BLINDED PLACEBO-CONTROLLED TRIAL
Reply to reviewers’ comments
The authors would like to thank the Reviewers for the constructive comments, criticism, and feedback. All the reviewers’ comments have been addressed. In addition, edits to the revised manuscript have been added as tracked changes in the text.
REVIEWER 1
The scientific paper "Effects of Photobiomodulation Therapy on Performance in Successive Time-to-Exhaustion Cycling Tests: A Randomized Double-Blinded Placebo-Controlled Trial" aimed to investigate the effects of photobiomodulation therapy (PBMT) on performance, oxygen uptake (VO2) kinetics, and lower limb muscle oxygenation during three successive time-to-exhaustion (TTE) in cyclists.
The manuscript contains an interesting clinical study. It can be considered that:
- In the abstract, in the first insertion of the acronym HHb insert the full meaning of the abbreviation.
Response: Thank you for your observation. Text has been modified as requested.
- Was the study registered at REBEC or did it use Consort? If yes, mention in the methodology.
Response: Thank you for your observation. Registration has not yet taken place. However, a document from the ethics and research committee of the Federal University of Rio Grande do Sul was requested for an opinion justifying that the manuscript in question is related to the doctoral project. In this way, we send the document to the editors/reviewers.
- Insert in the text clearly the inclusion and exclusion criteria of the participants.
Response: Thank you for your comment. Such suggestions are relevant and were included in the manuscript (Lines 112-118).
“The cyclists participating in the present study had ~6.5 years of regular training/competition and no history of lower limb muscle skeletal injuries. Inclusion criteria included cyclists with 18-30 years, with a competitive history and no history of musculoskeletal injuries in the lower limbs in the last two years. Exclusion criteria included chronic disease, smoking, metabolic disorders, use of steroids in the last six months, chronic disease, physical disabilities, smoking, and use of antibiotic drugs in the previous week.”
- Was the study developed during the COVID-19 pandemic? Did the cyclists wear masks that might alter the results? Please clarify.
Response: Thank you for your comment. The approval of the research ethics committee for the present project was carried out in 2014, while data collection of the present manuscript was carried out between the years 2015 and 2018. Therefore, the pandemic of COVID-19 had no effect on the results.
- In the manufacturers of the materials used, when they are from the United States, insert the state in which it is located. Example PBMT device.
Response: Thank you for your comment, all changes in text have been made.
- Preferably use the full name of the quadriceps muscle: quadriceps femoris.
Response: Thank you for your comment.
All changes in text have been made by inserting the full name “quadriceps femoris”.
- In the manufacturers of the materials used, insert the city in which it is located (manufacturer, city, country).
Response: Thank you for your comment, all changes in text have been made.
8) Figures should be colored for better visualization.
Response: Thank you for your comment. All figures have been changed to color.
9) In the footer of the tables, insert the abbreviations used in it and their meaning.
Response: Thank you for your comment. All tables and figures have been changed according to the comment.
10) For a clinical study, the number of bibliographic references should be close to 50 articles. Perhaps deepen the discussion on the relationship between PBMT and nerve injuries that affect the muscles (Ex: 10.3390/ph16050653).
Response: Thank you for your comment. Changes to the text were made to adjust the suggestions. Discussion (Lines: 380-388; 433-439); limitations (Lines: 453-456); Practical Application (Lines: 460-463) and in Conclusion.
11) In the conclusions, initially insert a brief context of the objectives and findings of the research, as well as ending with the benefits and clinical applicability.
Response: Thank you for your comment. The conclusion of the manuscript was adequate according to the commentary.
REVIEWER 2
1) There is no justification for the PBM methodology:
- why these particular zones are chosen;
- why 9 zones and not 3-4-5...18, for example?
- why these wavelengths and other parameters are chosen? Just because such a device was available?
- There is no such thing as dose (see publication)! Moskvin S.V. Response to: Optimization of Photobiomodulation Protocol for Chemotherapy-Induced Mucositis in Pediatric Patients (re: doi: 10.1089/photob.2019.4794) // Photobiomodulation, Photomedicine, and Laser Surgery. - 2020; 38 (11): 703. doi: 10.1089/photob.2020.4934.
Response: Thank you for your comment.
The application of photobiomodulation therapy (PBMT) in 9 zones of quadriceps femoris was due to two reasons: Initially because with these 9 zones (3 in each muscle) it practically covered the entire quadriceps femoris muscle group, as shown in figure 2. Furthermore, our previous study applied the same 9 zones and showed a positive effect on cycling performance at the three dosages tested (135, 270 and 405J/thigh); Lanferdini et al. (2018).
Regarding the choice of parameters, the configurations used in this equipment were similar to those adopted in the only previous study found and carried out by us in cyclists, which tested the three dosages mentioned above (Lanferdini et al. 2018). The fact that the same equipment used in the previous study was not used is due to the fact that it was unavailable at the time the present study was carried out. However, the application area with the use of this Chattanooga Cluster (33 diodes) is much larger than the previous study, covering almost entirely the quadriceps femoris with the 9 application zones of quadriceps femoris (Lanferdini et al. 2018). However, in this initial study, the effect of PBMT on successive time-to-exhaustion cycling tests was not verified. Furthermore, peripheral muscle oxygenation was not evaluated as in the present study.
However, Changes to the text were made to adjust the suggestions. Discussion (Lines: 380-388; 433-439); limitations (Lines: 453-456); Practical Application (Lines: 460-463) and in Conclusion.
Lanferdini, F.J.; Bini, R.R.; Baroni, B.M.; Klein, K.D.; Carpes, F.P.; Vaz, M.A. Improvement of performance and reduction of fatigue with low-level laser therapy in competitive cyclists. Int J Sports Physiol Perform. 2018, 13(1), 14-22.
Several other articles also report the dosages used (which are the basis for our primary study) that looked at the effect of dosage.
Albuquerque-Pontes GM, Vieira Rde P, Tomazoni SS, et al. Effect of pre-irradiation with different doses, wavelengths, and application intervals of low-level laser therapy on cytochrome c oxidase activity in intact skeletal muscle of rats. Lasers Med Sci. 2015;30(1):59–66.
Huang YY, Sharma SK, Carroll J, Hamblin MR. Biphasic dose response in low level light therapy—an update. Dose-Response. 2011;9(4):602–618.
Leal Junior EC, Lopes-Martins RA, Dalan F, et al. Effect of 655-nm low-level laser therapy on exercise-induced skeletal muscle fatigue in humans. Photomed Laser Surg. 2008;26(5):419–424.
Leal Junior EC, Lopes-Martins RA, Frigo L, et al. Effects of lowlevel laser therapy (LLLT) in the development of exercise-induced skeletal muscle fatigue and changes in biochemical markers related to postexercise recovery. J Orthop Sports Phys Ther. 2010;40(8): 524–532.
de Almeida P, Lopes-Martins RA, De Marchi T, et al. Red (660 nm) and infrared (830 nm) low-level laser therapy in skeletal muscle fatigue in humans: what is better? Lasers Med Sci. 2012;27(2):453–458.
Miranda EF, Leal-Junior EC, Marchetti PH, Dal Corso S. Acute effects of light emitting diodes therapy (LEDT) in muscle function during isometric exercise in patients with chronic obstructive pulmonary disease: preliminary results of a randomized controlled trial. Lasers Med Sci. 2014;29(1):359–365.
Baroni BM, Leal Junior EC, Geremia JM, Diefenthaeler F, Vaz MA. Effect of light-emitting diodes therapy (LEDT) on knee extensor muscle fatigue. Photomed Laser Surg. 2010a;28(5):653–658.
Baroni BM, Leal Junior EC, De Marchi T, Lopes LA, Salvador M, Vaz MA. Low level laser therapy before eccentric exercise reduces muscle damage markers in humans. Eur J Appl Physiol. 2010b;110(4):789–796.
De Marchi T, Leal Junior EC, Bortoli C, Tomazoni SS, LopesMartins RA, Salvador M. Low-level laser therapy (LLLT) in human progressive-intensity running: effects on exercise performance, skeletal muscle status, and oxidative stress. Lasers Med Sci. 2012;27(1):231–236.
Leal Junior EC, Lopes-Martins RA, Baroni BM, et al. Effect of 830 nm low-level laser therapy applied before high-intensity exercises on skeletal muscle recovery in athletes. Lasers Med Sci. 2009a;24(6):857–863.
Leal Junior EC, Lopes-Martins RA, Baroni BM, et al. Comparison between single-diode low-level laser therapy (LLLT) and LED multidiode (cluster) therapy (LEDT) applications before high-intensity exercise. Photomed Laser Surg. 2009b;27(4):617–623.
da Silva Alves MA, Pinfildi CE, Neto LN, Lourenco RP, de Azevedo PH, Dourado VZ. Acute effects of low-level laser therapy on physiologic and electromyographic responses to the cardiopulmonary exercise testing in healthy untrained adults. Lasers Med Sci. 2014;29(6):1945–1951.
Leal-Junior EC, Vanin AA, Miranda EF, de Carvalho Pde T, Dal Corso S, Bjordal JM. Effect of phototherapy (low-level laser therapy and light-emitting diode therapy) on exercise performance and markers of exercise recovery: a systematic review with meta-analysis. Lasers Med Sci. 2015;30(2):925–939.
Antonialli FC, De Marchi T, Tomazoni SS, et al. Phototherapy in skeletal muscle performance and recovery after exercise: effect of combination of super-pulsed laser and light-emitting diodes. Lasers Med Sci. 2014;29(6):1967–1976.
Dellagrana, R. A., Rossato, M., Sakugawa, R. L., Baroni, B. M., & Diefenthaeler, F. (2018a). Photobiomodulation therapy on physiological and performance parameters during running tests: Dose-response effects. The Journal of Strength and Conditioning Research, 32(10), 2807–2815.
Dellagrana, R. A., Rossato, M., Sakugawa, R. L., Lazzari, C. D., Baroni, B. M., & Diefenthaeler, F. (2018b). Dose-response effect of photobiomodulation therapy on neuromuscular economy during submaximal running. Lasers in Medical Science, 33(2), 329–336.
In addition, the text has been changed to adapt the suggestions and limitations of the study according to the suggestions.
2) An important aspect such as the corresponding changes in indicators outside the illumination zone, in other muscles, is not considered.
Response: Thank you for your comment.
We considered the possibility of treating other muscles with PBMT (Gluteus Maximus and Plantar Flexors). However, as mentioned in the previous comment. The quadriceps femoris is the most important muscle group in pedaling. Furthermore, the application in other muscles would be a problem, as it would take longer to apply and consequently could lose the effect of PBMT, as well as increase the interval between the maximum successive time-to-exhaustion tests.
However, in order to contemplate the request, we added to the limitations the fact that we did not apply PBMT in these muscle groups to help propel the crank cycle (lines 453-456). Furthermore, we changed the methods to justify the application on the quadriceps only (lines 174-178).
3) The fact that there is a direct effect of PBM on blood during out-of-muscle illumination is ignored (see publications below). Focusing attention only on muscles leads to erroneous conclusions regarding the mechanisms of biomodulating action of PBM.
Moskvin S.V., Khadartsev A.A. Methods of effective low-level laser therapy in the treatment of patients with bronchial asthma // BioMedicine. - 2020; 10 (1): 1-20. doi: 10.37796/2211-8039.1000.
Moskvin S.V., Kochetkov A.V. Russian low level laser therapy techniques for brain disorders // Photobiomodulation in the Brain. Low-Level Laser (Light) Therapy in Neurology and Neuroscience / M.R. Hamblin, Y.-Y. Huang (Eds). - London: Academic Press is an imprint of Elsevier, 2019. - P. 545-572.
Response: Thank you for your comment.
Therefore, changes to the text were made to adjust the suggestions. Discussion (Lines: 380-388; 433-439); limitations (Lines: 453-456); Practical Application (Lines: 460-463) and in Conclusion.
REFERENCES
Coyle, E.F.; Feltner, M.E.; Kautz, S.A.; Hamilton, M.T.; Montain, S.J.; Baylor, A.M.; Abraham, L.D.; Petrek, G.W. Physiological and biomechanical factors associated with elite endurance cycling performance. Med. Sci. Sport. Exerc. 1991, 23, 93–107.
Diefenthaeler, F.; Coyle, E.F.; Bini, R.B.; Carpes, F.P.; Vaz, M.A. Muscle activity and pedal force profile of triathletes during cycling to exhaustion. Sports Biomech. 2012, 11, 10-19.
Kordi, M.; Folland, J.; Goodall, S.; Haralabidis, N.; Maden-Wilkinson, T.; Sarika Patel, T.; Leeder, J.; Barratt, P.; Howatson, G. Mechanical and morphological determinants of peak power output in elite cyclists. Scan. J. Med. Sci. Sport. 2020, 30, 227–237.
Bathini, M.; Raghushaker, C.R.; Mahato, K.K. The molecular mechanisms of action of photobiomodulation against neurodegenerative diseases: a systematic review. Cell. Mol. Neurobiol. 2022, 42, 955-971.
Hamblin, M.R. Mechanisms and applications of the anti-inflammatory effects of photobiomodulation. AIMS. Biophys. 2017, 4, 337-361.
Moskvin, S.V.; Khadartsev, A.A. Methods of effective low-level laser therapy in the treatment of patients with bronchial asthma (literature review). Biomedicine. 2020, 28, 10, 1-20.
Dutra, Y.M.; Malta, E.S.; Elias, A.S.; Broatch, J.R.; Zagatto, A.M. Deconstructing the ergogenic effects of photobiomodulation: a systematic review and meta‑analysis of its efficacy in improving mode‑specific exercise performance in humans. Sports Med. 2022, 52, 2733-2757.
Elmer, S.J.; Barratt, P.R.; Korff, T.; Martin, J.C. Joint-specific power production during submaximal and maximal cycling. Med. Sci. Sport. Exerc. 2011, 43, 1940–1947.

Round 2
Reviewer 1 Report
-/-
Moderate editing of English language required
Author Response
Dear Rattikan Pancharoen
Editor of International Journal of Environmental Research and Public Health - MDPI
Reply to Reviewers’ Comments
Manuscript ID: ijerph-2438366
Article type: Original Research
Manuscript title: EFFECTS OF PHOTOBIOMODULATION THERAPY ON PERFORMANCE IN SUCCESSIVE TIME-TO-EXHAUSTION CYCLING TESTS: A RANDOMIZED DOUBLE-BLINDED PLACEBO-CONTROLLED TRIAL
Journal: International Journal of Environmental Research and Public Health
Reply to reviewers’ comments
The authors would like to thank the Reviewers for the constructive comments, criticism, and feedback. All the reviewers’ comments have been addressed. In addition, edits to the revised manuscript have been added as tracked changes in the text.
REVIEWER 1
The scientific paper "Effects of Photobiomodulation Therapy on Performance in Successive Time-to-Exhaustion Cycling Tests: A Randomized Double-Blinded Placebo-Controlled Trial" aimed to investigate the effects of photobiomodulation therapy (PBMT) on performance, oxygen uptake (VO2) kinetics, and lower limb muscle oxygenation during three successive time-to-exhaustion (TTE) in cyclists.
The manuscript contains an interesting clinical study. It can be considered that:
- In the abstract, in the first insertion of the acronym HHb insert the full meaning of the abbreviation.
Response: Thank you for your observation. Text has been modified as requested.
- Was the study registered at REBEC or did it use Consort? If yes, mention in the methodology.
Response: Thank you for your observation. Registration has not yet taken place. However, a document from the ethics and research committee of the Federal University of Rio Grande do Sul was requested for an opinion justifying that the manuscript in question is related to the doctoral project. In this way, we send the document to the editors/reviewers.
- Insert in the text clearly the inclusion and exclusion criteria of the participants.
Response: Thank you for your comment. Such suggestions are relevant and were included in the manuscript (Lines 112-118).
“The cyclists participating in the present study had ~6.5 years of regular training/competition and no history of lower limb muscle skeletal injuries. Inclusion criteria included cyclists with 18-30 years, with a competitive history and no history of musculoskeletal injuries in the lower limbs in the last two years. Exclusion criteria included chronic disease, smoking, metabolic disorders, use of steroids in the last six months, chronic disease, physical disabilities, smoking, and use of antibiotic drugs in the previous week.”
- Was the study developed during the COVID-19 pandemic? Did the cyclists wear masks that might alter the results? Please clarify.
Response: Thank you for your comment. The approval of the research ethics committee for the present project was carried out in 2014, while data collection of the present manuscript was carried out between the years 2015 and 2018. Therefore, the pandemic of COVID-19 had no effect on the results.
- In the manufacturers of the materials used, when they are from the United States, insert the state in which it is located. Example PBMT device.
Response: Thank you for your comment, all changes in text have been made.
- Preferably use the full name of the quadriceps muscle: quadriceps femoris.
Response: Thank you for your comment.
All changes in text have been made by inserting the full name “quadriceps femoris”.
- In the manufacturers of the materials used, insert the city in which it is located (manufacturer, city, country).
Response: Thank you for your comment, all changes in text have been made.
8) Figures should be colored for better visualization.
Response: Thank you for your comment. All figures have been changed to color.
9) In the footer of the tables, insert the abbreviations used in it and their meaning.
Response: Thank you for your comment. All tables and figures have been changed according to the comment.
10) For a clinical study, the number of bibliographic references should be close to 50 articles. Perhaps deepen the discussion on the relationship between PBMT and nerve injuries that affect the muscles (Ex: 10.3390/ph16050653).
Response: Thank you for your comment. Changes to the text were made to adjust the suggestions. Discussion (Lines: 380-388; 433-439); limitations (Lines: 453-456); Practical Application (Lines: 460-463) and in Conclusion.
11) In the conclusions, initially insert a brief context of the objectives and findings of the research, as well as ending with the benefits and clinical applicability.
Response: Thank you for your comment. The conclusion of the manuscript was adequate according to the commentary.
REVIEWER 2
1) There is no justification for the PBM methodology:
- why these particular zones are chosen;
- why 9 zones and not 3-4-5...18, for example?
- why these wavelengths and other parameters are chosen? Just because such a device was available?
- There is no such thing as dose (see publication)! Moskvin S.V. Response to: Optimization of Photobiomodulation Protocol for Chemotherapy-Induced Mucositis in Pediatric Patients (re: doi: 10.1089/photob.2019.4794) // Photobiomodulation, Photomedicine, and Laser Surgery. - 2020; 38 (11): 703. doi: 10.1089/photob.2020.4934.
Response: Thank you for your comment.
The application of photobiomodulation therapy (PBMT) in 9 zones of quadriceps femoris was due to two reasons: Initially because with these 9 zones (3 in each muscle) it practically covered the entire quadriceps femoris muscle group, as shown in figure 2. Furthermore, our previous study applied the same 9 zones and showed a positive effect on cycling performance at the three dosages tested (135, 270 and 405J/thigh); Lanferdini et al. (2018).
Regarding the choice of parameters, the configurations used in this equipment were similar to those adopted in the only previous study found and carried out by us in cyclists, which tested the three dosages mentioned above (Lanferdini et al. 2018). The fact that the same equipment used in the previous study was not used is due to the fact that it was unavailable at the time the present study was carried out. However, the application area with the use of this Chattanooga Cluster (33 diodes) is much larger than the previous study, covering almost entirely the quadriceps femoris with the 9 application zones of quadriceps femoris (Lanferdini et al. 2018). However, in this initial study, the effect of PBMT on successive time-to-exhaustion cycling tests was not verified. Furthermore, peripheral muscle oxygenation was not evaluated as in the present study.
However, Changes to the text were made to adjust the suggestions. Discussion (Lines: 380-388; 433-439); limitations (Lines: 453-456); Practical Application (Lines: 460-463) and in Conclusion.
Lanferdini, F.J.; Bini, R.R.; Baroni, B.M.; Klein, K.D.; Carpes, F.P.; Vaz, M.A. Improvement of performance and reduction of fatigue with low-level laser therapy in competitive cyclists. Int J Sports Physiol Perform. 2018, 13(1), 14-22.
Several other articles also report the dosages used (which are the basis for our primary study) that looked at the effect of dosage.
Albuquerque-Pontes GM, Vieira Rde P, Tomazoni SS, et al. Effect of pre-irradiation with different doses, wavelengths, and application intervals of low-level laser therapy on cytochrome c oxidase activity in intact skeletal muscle of rats. Lasers Med Sci. 2015;30(1):59–66.
Huang YY, Sharma SK, Carroll J, Hamblin MR. Biphasic dose response in low level light therapy—an update. Dose-Response. 2011;9(4):602–618.
Leal Junior EC, Lopes-Martins RA, Dalan F, et al. Effect of 655-nm low-level laser therapy on exercise-induced skeletal muscle fatigue in humans. Photomed Laser Surg. 2008;26(5):419–424.
Leal Junior EC, Lopes-Martins RA, Frigo L, et al. Effects of lowlevel laser therapy (LLLT) in the development of exercise-induced skeletal muscle fatigue and changes in biochemical markers related to postexercise recovery. J Orthop Sports Phys Ther. 2010;40(8): 524–532.
de Almeida P, Lopes-Martins RA, De Marchi T, et al. Red (660 nm) and infrared (830 nm) low-level laser therapy in skeletal muscle fatigue in humans: what is better? Lasers Med Sci. 2012;27(2):453–458.
Miranda EF, Leal-Junior EC, Marchetti PH, Dal Corso S. Acute effects of light emitting diodes therapy (LEDT) in muscle function during isometric exercise in patients with chronic obstructive pulmonary disease: preliminary results of a randomized controlled trial. Lasers Med Sci. 2014;29(1):359–365.
Baroni BM, Leal Junior EC, Geremia JM, Diefenthaeler F, Vaz MA. Effect of light-emitting diodes therapy (LEDT) on knee extensor muscle fatigue. Photomed Laser Surg. 2010a;28(5):653–658.
Baroni BM, Leal Junior EC, De Marchi T, Lopes LA, Salvador M, Vaz MA. Low level laser therapy before eccentric exercise reduces muscle damage markers in humans. Eur J Appl Physiol. 2010b;110(4):789–796.
De Marchi T, Leal Junior EC, Bortoli C, Tomazoni SS, LopesMartins RA, Salvador M. Low-level laser therapy (LLLT) in human progressive-intensity running: effects on exercise performance, skeletal muscle status, and oxidative stress. Lasers Med Sci. 2012;27(1):231–236.
Leal Junior EC, Lopes-Martins RA, Baroni BM, et al. Effect of 830 nm low-level laser therapy applied before high-intensity exercises on skeletal muscle recovery in athletes. Lasers Med Sci. 2009a;24(6):857–863.
Leal Junior EC, Lopes-Martins RA, Baroni BM, et al. Comparison between single-diode low-level laser therapy (LLLT) and LED multidiode (cluster) therapy (LEDT) applications before high-intensity exercise. Photomed Laser Surg. 2009b;27(4):617–623.
da Silva Alves MA, Pinfildi CE, Neto LN, Lourenco RP, de Azevedo PH, Dourado VZ. Acute effects of low-level laser therapy on physiologic and electromyographic responses to the cardiopulmonary exercise testing in healthy untrained adults. Lasers Med Sci. 2014;29(6):1945–1951.
Leal-Junior EC, Vanin AA, Miranda EF, de Carvalho Pde T, Dal Corso S, Bjordal JM. Effect of phototherapy (low-level laser therapy and light-emitting diode therapy) on exercise performance and markers of exercise recovery: a systematic review with meta-analysis. Lasers Med Sci. 2015;30(2):925–939.
Antonialli FC, De Marchi T, Tomazoni SS, et al. Phototherapy in skeletal muscle performance and recovery after exercise: effect of combination of super-pulsed laser and light-emitting diodes. Lasers Med Sci. 2014;29(6):1967–1976.
Dellagrana, R. A., Rossato, M., Sakugawa, R. L., Baroni, B. M., & Diefenthaeler, F. (2018a). Photobiomodulation therapy on physiological and performance parameters during running tests: Dose-response effects. The Journal of Strength and Conditioning Research, 32(10), 2807–2815.
Dellagrana, R. A., Rossato, M., Sakugawa, R. L., Lazzari, C. D., Baroni, B. M., & Diefenthaeler, F. (2018b). Dose-response effect of photobiomodulation therapy on neuromuscular economy during submaximal running. Lasers in Medical Science, 33(2), 329–336.
In addition, the text has been changed to adapt the suggestions and limitations of the study according to the suggestions.
2) An important aspect such as the corresponding changes in indicators outside the illumination zone, in other muscles, is not considered.
Response: Thank you for your comment.
We considered the possibility of treating other muscles with PBMT (Gluteus Maximus and Plantar Flexors). However, as mentioned in the previous comment. The quadriceps femoris is the most important muscle group in pedaling. Furthermore, the application in other muscles would be a problem, as it would take longer to apply and consequently could lose the effect of PBMT, as well as increase the interval between the maximum successive time-to-exhaustion tests.
However, in order to contemplate the request, we added to the limitations the fact that we did not apply PBMT in these muscle groups to help propel the crank cycle (lines 453-456). Furthermore, we changed the methods to justify the application on the quadriceps only (lines 174-178).
3) The fact that there is a direct effect of PBM on blood during out-of-muscle illumination is ignored (see publications below). Focusing attention only on muscles leads to erroneous conclusions regarding the mechanisms of biomodulating action of PBM.
Moskvin S.V., Khadartsev A.A. Methods of effective low-level laser therapy in the treatment of patients with bronchial asthma // BioMedicine. - 2020; 10 (1): 1-20. doi: 10.37796/2211-8039.1000.
Moskvin S.V., Kochetkov A.V. Russian low level laser therapy techniques for brain disorders // Photobiomodulation in the Brain. Low-Level Laser (Light) Therapy in Neurology and Neuroscience / M.R. Hamblin, Y.-Y. Huang (Eds). - London: Academic Press is an imprint of Elsevier, 2019. - P. 545-572.
Response: Thank you for your comment.
Therefore, changes to the text were made to adjust the suggestions. Discussion (Lines: 380-388; 433-439); limitations (Lines: 453-456); Practical Application (Lines: 460-463) and in Conclusion.
REFERENCES
Coyle, E.F.; Feltner, M.E.; Kautz, S.A.; Hamilton, M.T.; Montain, S.J.; Baylor, A.M.; Abraham, L.D.; Petrek, G.W. Physiological and biomechanical factors associated with elite endurance cycling performance. Med. Sci. Sport. Exerc. 1991, 23, 93–107.
Diefenthaeler, F.; Coyle, E.F.; Bini, R.B.; Carpes, F.P.; Vaz, M.A. Muscle activity and pedal force profile of triathletes during cycling to exhaustion. Sports Biomech. 2012, 11, 10-19.
Kordi, M.; Folland, J.; Goodall, S.; Haralabidis, N.; Maden-Wilkinson, T.; Sarika Patel, T.; Leeder, J.; Barratt, P.; Howatson, G. Mechanical and morphological determinants of peak power output in elite cyclists. Scan. J. Med. Sci. Sport. 2020, 30, 227–237.
Bathini, M.; Raghushaker, C.R.; Mahato, K.K. The molecular mechanisms of action of photobiomodulation against neurodegenerative diseases: a systematic review. Cell. Mol. Neurobiol. 2022, 42, 955-971.
Hamblin, M.R. Mechanisms and applications of the anti-inflammatory effects of photobiomodulation. AIMS. Biophys. 2017, 4, 337-361.
Moskvin, S.V.; Khadartsev, A.A. Methods of effective low-level laser therapy in the treatment of patients with bronchial asthma (literature review). Biomedicine. 2020, 28, 10, 1-20.
Dutra, Y.M.; Malta, E.S.; Elias, A.S.; Broatch, J.R.; Zagatto, A.M. Deconstructing the ergogenic effects of photobiomodulation: a systematic review and meta‑analysis of its efficacy in improving mode‑specific exercise performance in humans. Sports Med. 2022, 52, 2733-2757.
Elmer, S.J.; Barratt, P.R.; Korff, T.; Martin, J.C. Joint-specific power production during submaximal and maximal cycling. Med. Sci. Sport. Exerc. 2011, 43, 1940–1947.

Round 3
Reviewer 1 Report
No Comments. Thanks
Minor editing